# Accelerating Reinforcement Learning through GPU Atari Emulation

**Steven Dalton**,* **Iuri Frosio**\*
NVIDIA, USA
{sdalton,ifrosio}@nvidia.com

## Abstract

We introduce CuLE (CUDA Learning Environment), a CUDA port of the Atari Learning Environment (ALE) which is used for the development of deep reinforcement algorithms. CuLE overcomes many limitations of existing CPU-based emulators and scales naturally to multiple GPUs. It leverages GPU parallelization to run thousands of games simultaneously and it renders frames directly on the GPU, to avoid the bottleneck arising from the limited CPU-GPU communication bandwidth. CuLE generates up to 155M frames per hour on a single GPU, a finding previously achieved only through a cluster of CPUs. Beyond highlighting the differences between CPU and GPU emulators in the context of reinforcement learning, we show how to leverage the high throughput of CuLE by effective batching of the training data, and show accelerated convergence for A2C+V-trace. CuLE is available at `https://github.com/NVlabs/cule`.

## 1 Introduction

Initially triggered by the success of DQN [13], research in Deep Reinforcement Learning (DRL) has grown in popularity in the last years [10, 12, 13], leading to intelligent agents that solve non-trivial tasks in complex environments. But DRL also soon proved to be a challenging computational problem, especially if one wants to achieve peak performance on modern architectures.

Traditional DRL training focuses on CPU environments that execute a set of actions $\{a_{t-1}\}$ at time $t-1$, and produce observable states $\{s_t\}$ and rewards $\{r_t\}$. Environment data is then processed by a Deep Neural Network (DNN) on the GPU to select the next action, $\{a_t\}$, which is copied back to the CPU. This sequence of operations defines the *inference path*, whose main aim is to generate training data. A training buffer on the GPU stores the states generated on the *inference path*; this is periodically used to update the DNN's weights $\theta$, according to the training rule of the DRL algorithm (*training path*). A computationally efficient DRL system should balance the data generation and training processes, while minimizing the communication overhead along the *inference path* and consuming, along the *training path*, as many data per second as possible [1, 2]. The solution to this problem is however non-trivial and many DRL implementations do not leverage the full computational potential of modern systems [19].

We focus our attention on the *inference path* and move from the traditional CPU implementation of the Atari Learning Environment (ALE), a set of Atari 2600 games that emerged as an excellent DRL benchmark [3, 11]. We show that significant performance bottlenecks stem from CPU-based environment emulation because the CPU cannot run a large set of environments simultaneously and the CPU-GPU communication bandwidth is limited. To both investigate and mitigate these limitations we introduce CuLE (CUDA Learning Environment), a DRL library containing a CUDA enabled

---

Table 1: Average training times, raw frames to reach convergence, FPS, and computational resources of existing accelerated DRL schemes, compared to CuLE. Data from [8]; FPS are taken from the corresponding papers, if available, and measured on the entire Atari suite for CuLE.

| Algorithm | Time | Frames | FPS | Resources |
|---|---|---|---|---|
| Ape-X DQN [8] | 5 days | 22,800M | 50K | 376 cores, 1 GPU |
| Rainbow [7] | 10 days | 200M | — | 1 GPU |
| Distributional (C51) [4] | 10 days | 200M | — | 1 GPU |
| A3C [12] | 4 days | — | 2K | 16 cores |
| GA3C [1, 2] | 1 day | — | 8K | 16 cores, 1 GPU |
| Prioritized Dueling [22] | 9.5 days | 200M | — | 1 GPU |
| DQN [13] | 9.5 days | 200M | — | 1 GPU |
| Gorila DQN [14] | 4 days | — | — | > 100 cores |
| Unreal [9] | — | 250M | — | 16 cores |
| Stooke (A2C / DQN) [19] | hours | 200M | 35K | 40 CPUs, 8 GPUs (DGX-1) |
| IMPALA (A2C + V-Trace) [6] | mins/hours | 200M | 250K | 100-200 cores, 1 GPU |
| CuLE (*emulation only*) | N/A | N/A | 41K-155K | System I (1 GPU) |
| CuLE (*inference only*, A2C, single batch) | N/A | N/A | 39K-125K | System I (1 GPU) |
| CuLE (*training*, A2C + V-trace, multiple batches) | 1 hour | 200M | 26K-68K | System I (1 GPU) |
| CuLE (*training*, A2C + V-trace, multiple batches)* | mins | 200M | 142-187K | System III (4 GPUs) |

*FPS measured on Asterix, Assault, MsPacman, and Pong.

Table 2: Systems used for experiments.

| System | Intel CPU | NVIDIA GPU |
|---|---|---|
| I | 12-core Core i7-5930K @3.50GHz | Titan V |
| II | 6-core Core i7-8086K @5GHz | Tesla V100 |
| III | 20-core Core E5-2698 v4 @2.20GHz $\times$ 2 | Tesla V100 $\times$ 8, NVLink |

Atari 2600 emulator that renders frames directly in GPU memory, avoids off-chip communication and achieves high GPU utilization by processing thousands of environments in parallel—something so far achievable only through large and costly distributed systems. Compared to the traditional CPU-based approach, GPU emulation improves the utilization of the computational resources: CuLE on a single GPU generates more Frames Per Second[2] (FPS) on the *inference path* (between 39K and 125K, depending on the game, see Table 1) compared to its CPU counterpart (between 12.5K and 19.8K). CuLE's throughput is of the same order of magnitude of much larger distributed systems, like IMPALA [6] or a DGX-1 [20], which eventually leads to a significant reduction in the wall clock training time and therefore to an immediate practical advantage [18, 21, 5] for the researchers working in this field. Beyond offering CuLE (`https://github.com/NVlabs/cule`) as a tool for research in the DRL field, our contribution can be summarized as follow:

**(1)** We identify common computational bottlenecks in several DRL implementations that prevent effective utilization of high throughput compute units and effective scaling to distributed systems.

**(2)** We introduce an effective batching strategy for large environment sets, that allows leveraging the high throughput generated by CuLE to quickly reach convergence with A2C+V-trace [6], and show effective scaling on multiple GPUs. This leads to the consumption of 26-68K FPS along the *training path* on a single GPU, and up to 187K FPS using four GPUs, comparable (Table 1) to those achieved by large clusters [20, 6].

**(3)** We analyze advantages and limitations of GPU emulation with CuLE in DRL, including the effect of thread divergence and of the lower (compared to CPU) number of instructions per second per thread, and hope that our insights may be of value for the development of efficient DRL systems.

## 2 Related Work

The wall clock convergence time of a DRL algorithm is determined by two main factors: its *sample efficiency* and the *computational efficiency* of its implementation. Here we analyze the sample and computational efficiency of different DRL algorithms, in connection with their implementation.

We first divide DRL algorithms into policy gradient and Q-value methods, as in [19]. Q-learning optimizes the error on the estimated action values as a proxy for policy optimization, whereas policy gradient methods directly learn the relation between a state, $s_t$, and the optimal action, $a_t$; since at each update they follow, by definition, the gradient with respect to the policy itself, they improve the policy more efficiently. Policy methods are also considered more general, e.g. they can handle continuous actions easily. Also the on- or off-policy nature of an algorithm profoundly affects both its sample and computational efficiency. Off-policy methods allow re-using experiences multiple times, which directly improves the sample efficiency; additionally, old data stored in GPU memory can be used to continuously update the DNN, leading to high GPU utilization without saturating the *inference path*. The replay buffer has a positive effect on the stability of learning as well [13]. On-policy algorithms saturate the *inference path* more easily, as frames have to be generated on-the-fly using the current policy and moved from the CPU to the GPU for processing with the DNN. On-policy updates are generally effective but they are also more prone to fall into local minima because of noise, especially if the number of environment is small — this is the reason why on-policy algorithms largely benefit (in term of stability) from a significant increase of the number of environments.

Policy gradient algorithms are often on-policy: their efficient update strategy is counterbalanced by the bottlenecks in the *inference path* and competition for the use of the GPU along the *inference* and *training path* at the same time. Acceleration by scaling to a distributed system is possible but inefficient in this case: in IMPALA [6] a cluster with hundreds of CPU cores is needed to accelerate A2C, while training is desynchronized to hide latency. As a consequence, the algorithm becomes off-policy, and V-trace was introduced to deal with off-policy data (see details in the Appendix). Acceleration on a DGX-1 has also been demonstrated for A2C and PPO, using large batch sizes to increase the GPU occupancy, and asynchronous distributed models that hide latency, but require periodic updates to remain synchronized [19] and overall achieves sublinear scaling with the number of GPUs.

During the review process, we came to know about *Sample Factory* [17], a high-throughput training system that, like CuLE, is designed to optimize the efficiency and resource utilization of reinforcement learning algorithms on a single machine. In contrast with CuLE, *Sample Factory* uses CPU simulation and a novel GPU-based sampler; like in the case of CuLE and other high throughput systems [6, 19], off-policy correction is then needed to process the large amount of generated data. Studying in detail the commonalities and complementary aspects of CuLE and *Sample Factory* may easily lead to further developments towards the creation of computationally efficient reinforcement learning systems.

## 3 CUDA Learning Environment (CuLE)

In CuLE, we emulate the functionality of many Atari emulators in parallel using the CUDA programming model, where a sequential host program executes parallel programs, known as kernels, on the GPU. In a trivial mapping of the Atari emulator to CUDA, a single thread emulates both the Atari CPU and TIA processors to read code from the ROM, advance the game state, and update pixels in the framebuffer. However, the contrasting nature of the code execution, dominated by reading from RAM/ROM, and frame rendering tasks, writing hundreds of pixels to the framebuffer, poses a serious issue in terms of performance, such as thread divergence and an imbalanced number of kernel resources, such as registers, required by each task. To mitigate these issues, CuLE uses two CUDA kernels: the first one first loads data from GPU global memory, where we store the state of each emulator, and the 128 bytes RAM data; it also reads the ROM instructions from constant memory then executes and stores the updated game state into global memory. It is important to note that this first kernel does not execute the TIA instructions read from the ROM, but copies them into a TIA instruction buffer, which we implemented to decouple the execution of the CPU and TIA instructions in CuLE. The second CuLE kernel emulates the functionality of the TIA processor: it first reads the instructions stored in the TIA instruction buffer, executes them to update the TIA registers, and renders the $160 \times 210$ output framebuffer in global GPU memory. Despite processing

the TIA instructions twice this implementation has several advantages over the single-kernel trivial implementation. First of all, the requirements, in terms of registers per thread, and the chance of having divergent code are different for the Atari CPU and TIA kernels, and the use of different kernels improves GPU utilization. A second advantage that we exploit is that not all frames are rendered in ALE: the input of the RL algorithm is the pixelwise maximum between the last two frames in a sequence of four, so we can avoid calling the TIA kernel when rendering of the screen is not needed. A last advantage, not exploited in our implementation yet, is that the TIA kernel may be scheduled on the GPU with more than one thread per game, as rendering of diverse rows on the screen is indeed a parallel operation - we leave this optimization for future developments of CuLE.

To better fit our execution model, our game reset strategy is also different from the one in the existing CPU emulators, where 64 startup frames are executed at the end of each episode. Furthermore, wrapper interfaces for RL, such as ALE, randomly execute an additional number of frames (up to 30) to introduce randomness into the initial state. This results in up to 94 frames to reset a game, which may cause massive divergence between thousands of emulators executing in a SIMD fashion on the GPU. To address this issue, we generate and store a cache of random initial states (30 by default) when a set of environments are initialized in CuLE. At the end of an episode, each emulator randomly selects one of the cached states as a seed and copies it into the terminal emulator state.

Some of the choices made for the implementation of CuLE are informed by ease of debugging, like associating one state update kernel to one environment, or need for flexibility, like emulating the Atari console instead of directly writing CUDA code for

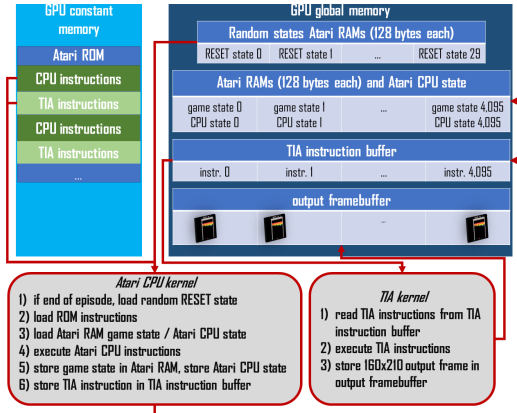

Figure 1: Our CUDA-based Atari emulator uses an *Atari CPU kernel* to emulate the functioning of the Atari CPU and advance the game state, and a second *TIA kernel* to emulate the TIA and render frames directly in GPU memory. For episode resetting we generate and store a cache of random initial states. Massive parallelization on GPU threads allows the parallel emulation of thousands of Atari games.

each Atari game. A 1-to-1 mapping between threads and emulators is not the most computationally efficient way to run Atari games on a GPU, but it makes the implementation relatively straightforward and has the additional advantage that the same emulator code can be executed on the CPU for debugging and benchmarking (in the following, we will refer to this implementation as CuLE$_{\text{CPU}}$). Despite these issues, the computational advantage provided by CuLE over traditional CPU emulation remains significant.

## 4 Experiments

**Atari emulation** We measure the FPS under different conditions: we get an upper bound on the maximum achievable FPS in the *emulation only* case, when we emulate the environments and use a random policy to select actions. In the *inference only* case, we measure the FPS along the *inference path*: a policy DNN selects the actions, CPU-GPU data transfer occur for CPU emulators, while both emulation and DNN inference run on the GPU when CuLE is used. This is the maximum throughput achievable by off-policy algorithms, when data generation and consumption are decoupled and run on different devices. In the *training* case, the entire DRL system is at work: emulation, inference, and training may all run on the same GPU. This is representative of the case of on-policy algorithms, but the FPS are also affected by the computational cost of the specific DRL update algorithm; in our experiments we use a vanilla A2C [15], with N-step bootstrapping, and $N = 5$ as the baseline (for details of A2C and off-policy correction with V-trace, see the Appendix).

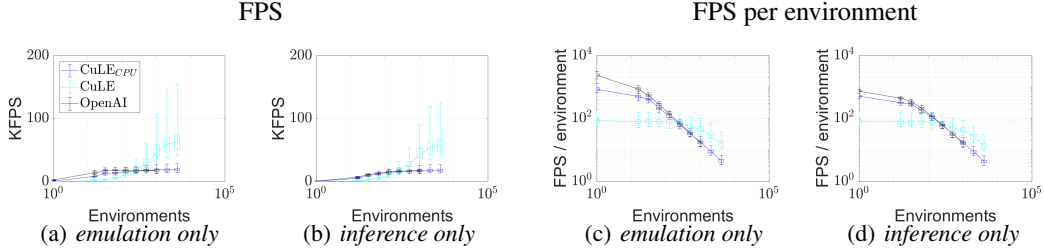

Figure 2: FPS and FPS / environment on System I in Table 2, for OpenAI Gym [15], CuLE$_{CPU}$, and CuLE, as a function of the number of environments, under different load conditions: *emulation only*, and *inference only*. The boxplots indicate the minimum, $25^{th}$, $50^{th}$, $75^{th}$ percentiles and maximum FPS, for the entire set of 57 Atari games.

Table 3: Training FPS, DNN's Update Per Second (UPS), time to reach a given score, and corresponding number of training frames for four Atari games, A2C+V-trace, and different configurations of the emulation engines, measured on System I in Table 2 (System III for the multi-GPU case). The best metric in each row is in bold.

| Engine | OpenAI Gym | | | | CuLE, 1 GPU | | | CuLE, 4 GPUs | Game |
|---|---|---|---|---|---|---|---|---|---|
| Envs | 120 | 120 | 120 | 1200 | 1200 | 1200 | 1200 | 1200×4 | |
| Batches | 1 | 5 | 20 | 20 | 1 | 5 | 20 | 20×4 | |
| N-steps | 5 | 5 | 20 | 20 | 5 | 5 | 20 | 20 | |
| SPU | 5 | 1 | 1 | 1 | 5 | 1 | 1 | 1 | |
| Training KFPS | 4.2 | 3.4 | 3.0 | 4.9 | 10.6 | 11.5 | 11.0 | **42.7** | |
| UPS | 7.0 | **28.3** | 24.7 | 4.1 | 1.8 | 9.6 | 9.1 | 8.9 | Assault |
| Time [mins] | 20.2 | — | 42.6 | 44.2 | 18.8 | 9.4 | 9.9 | **7.9** | |
| Training Mframes (for average score: 800) | **5.0** | — | 7.5 | 13.0 | 12.0 | 6.5 | 6.5 | 18.0 | |
| Training KFPS | 4.3 | 3.3 | 3.0 | 4.9 | 11.9 | 12.5 | 12.1 | **46.6** | |
| UPS | 7.1 | **27.9** | 24.8 | 4.1 | 2.0 | 10.4 | 10.0 | 9.7 | Asterix |
| Time [mins] | 8.1 | 35.2 | 14.4 | 27.1 | — | 14.0 | 3.4 | **2.5** | |
| Training Mframes (for average score: 1,000) | **2.0** | 7.0 | 2.5 | 8.0 | — | 10.5 | 2.5 | 7.0 | |
| Training KFPS | 4.0 | 3.3 | 2.8 | 4.8 | 9.0 | 9.6 | 9.2 | **35.5** | |
| UPS | 6.7 | **27.1** | 23.7 | 4.0 | 1.5 | 8.0 | 7.7 | 7.4 | MsPacman |
| Time [mins] | 16.6 | 20.5 | 14.7 | 12.4 | — | 6.9 | 11.8 | **2.4** | |
| Training Mframes (for average score: 1,500) | 4.0 | 4.0 | **2.5** | 3.5 | — | 4.0 | 6.5 | 3.0 | |
| Training KFPS | 4.3 | 3.4 | 3.0 | 4.8 | 10.5 | 11.2 | 10.6 | **41.7K** | |
| UPS | 7.2 | **28.1** | 24.9 | 4.0 | 1.8 | 9.3 | 8.9 | 8.7 | Pong |
| Time [mins] | 21.2 | 12.2 | 8.4 | 8.7 | — | 5.9 | 3.1 | **2.4** | |
| Training Mframes (for average score: 18) | 5.5 | 2.5 | **1.5** | 2.5 | — | 4.0 | 2.0 | 6.0 | |

Figs. 2(a)-2(b) show the FPS generated by OpenAI Gym, CuLE$_{CPU}$, and CuLE, on the entire set of Atari games, as a function of the number of environments. In the *emulation only* case, CPU emulation is more efficient for a number of environments up to 128, when the GPU computational power is not leveraged because of the low occupancy. For a larger number of environments, CuLE significantly overcomes OpenAI Gym, for which FPS are mostly stable for 64 environments or more, indicating that the CPU is saturated: the ratio between the median FPS generated by CuLE with 4096 environment (64K) and the peak FPS for OpenAI Gym (18K) is $3.56\times$. In the *inference only* case there are two additional overheads: CPU-GPU communication (to transfer observations), and DNN inference on the GPU. Consequently, CPU emulators achieve a lower FPS in *inference only* when compared to *emulation only*; the effects of the overheads is more evident for a small number of environments, while the FPS slightly increase with the number of environments without reaching the *emulation only* FPS. CuLE's FPS are also lower for *inference only*, because of the latency introduced by DNN inference, but the FPS grow with the number of environments, suggesting that the computational capability of the GPU is still far from being saturated.

**Factors affecting the FPS** Figs. 2(a)-2(b) shows that the throughput varies dramatically across games: 4096 CuLE$_{CPU}$ environments run at 27K FPS on Riverraid, but only 14K FPS for Boxing: a $1.93\times$ difference, explained by the different complexity of the ROM code of each game. The ratio

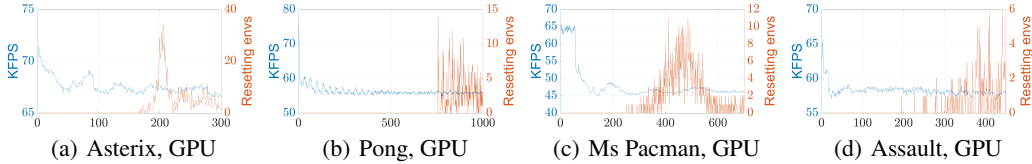

(a) Asterix, GPU          (b) Pong, GPU          (c) Ms Pacman, GPU          (d) Assault, GPU

Figure 3: FPS as a function of the environment step, measured on System I in Table 2 for *emulation only* on four Atari games, 512 environments, for CuLE; each panel also shows the number of resetting environments. FPS is higher at the beginning, when all environments are in similar states and thread divergence within warps is minimized; after some steps, correlation is lost, FPS decreases and stabilizes. Minor oscillations in FPS are possibly associated to more or less computational demanding phases in the emulation of the environments (e.g., when a goal is scored in Pong).

between the maximum and minimum FPS is amplified in the case of GPU emulation: Riverraid runs in *emulation only* at 155K FPS when emulated by CuLE and 4096 environments, while UpNDown runs at 41K FPS —a $3.78\times$ ratio.

To better highlight the impact of thread divergence on throughput, we measure the FPS for CuLE, *emulation only*, 512 environments, and four games (Fig. 3). All the environments share the same initial state, but random action selection leads them to diverge after some steps. Each environment resets at the end of an episode. The FPS is maximum at the very beginning, when all the environments are in similar states and the chance to execute the same instruction in all the threads is high. When they move towards different states, code divergence negatively impacts the FPS, until it reaches an asymptotic value. This effect is present in all games and particularly evident for MsPacman in Fig. 3; it is not present in CPU emulation (see Appendix). Although divergence can reduce FPS by 30% in the worst case, this has to be compared with case of complete divergence within each thread and for each instruction, which would yield $1/32 \simeq 3\%$ of the peak performances. Minor oscillations of the FPS are also visible especially for games with a repetitive pattern (e.g. Pong), where different environments can be more or less correlated with a typical oscillation frequency.

**Performances during training**   Fig. 4 compares the FPS generated by different emulation engines on a specific game (Assault)[3], for different load conditions, including the *training* case, and number of environments. As expected, when the entire *training path* is at work, the FPS decreases even further. However, for CPU emulators, the difference between FPS in the *inference only* and *training* cases decreases when the number of environments increases, as the system is bounded by the CPU computational capability and CPU-GPU communication bandwidth. In the case of the CPU, scaling to multiple GPUs would be ineffective for on-policy algorithms, such GA3C [1, 2], or sub-optimal, in the case of distributed systems [6, 19]. On the other hand, the difference between *inference only* and *training* FPS increases with the number of environments for CuLE, because of the additional

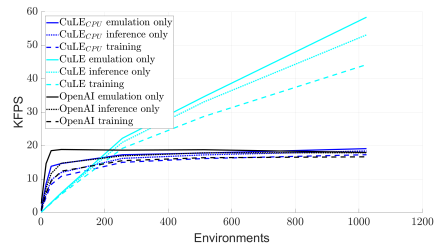

Figure 4: FPS generated by different emulation engines on System I in Table 2 for Assault, as a function of the number of environments, and different load conditions for A2C with N-step bootstrapping, $N = 5$).

training overhead on the GPU. The potential speed-up provided by CuLE for vanilla A2C and Assault in Fig. 4 is $2.53\times$ for 1,024 environments, but the system is bounded by the GPU computational power; as a consequence, better batching strategies that reduce the training computational overhead as well as scaling to multiple GPUs are effective to further increase the speed-up ratio, as demonstrated later in this Section.

When data generation and training can be decoupled, like for off-policy algorithms, training can be easily moved to a different GPU and the *inference path* can be used at maximum speed. The potential speed-up provided by CuLE for off-policy algorithms is then given by the ratio between the *inference only* median FPS for CuLE (56K) and CuLE$_{CPU}$ (18K), which is $3.11\times$ for 4,096 environments.

Furthermore, since the FPS remains flat for CPU emulation, the advantage of CuLE amplifies (for both on- and off-policy methods) with the number of environments.

**Frames per second per environment** Fig. 2(c)-2(d) show the FPS / environment for different emulation engines on System I, as a function of the number of environments. For 128 environments or fewer, CPU emulators generate frames at a higher rate (compared to CuLE), because CPUs are optimized for low latency, and execute a high number of instructions per second per thread. However, the FPS / environment decrease with the number of environments, that have to share the same CPU cores. Instead, the GPU architecture maximizes the throughput and has a lower number of instructions per second per thread. As a consequence, the FPS / environment is smaller (compared to CPU emulation) for a small number of environments, but they are almost constant up to 512 environments, and starts decreasing only after this point. In practice, CuLE environments provide an efficient means of training with a diverse set of data and collect large statistics about the rewards experienced by numerous agents, and consequently lowering the variance of the value estimate. On the other hand, samples are collected less efficiently in the temporal domain, which may worsen the bias on the estimate of the value function by preventing the use of large N in N-step bootstrapping. The last paragraph of this Section shows how to leverage the high throughput generated by CuLE, considering these peculiarities.

**Memory limitations** Emulating a massively large number of environments can be problematic considering the relatively small amount of GPU DRAM. Our PyTorch [16] implementation of A2C requires each environment to store 4 84x84 frames, plus some additional variables for the emulator state. For 16K environments this translates into 1GB of memory, but the primary issue is the combined memory pressure to store the DNN with 4M parameters and the meta-data during training, including the past states: training with 16K environments easily exhausts the DRAM on a single GPU (while training on multiple GPUs increases the amount of available RAM). Since we did not implement any data compression scheme as in [8], we constrain our training configuration to fewer than 5K environments, but peak performance in terms of FPS would be achieved for a higher number of environments - this is left as a possible future improvement.

**A2C** We analyze in detail the case of A2C with CuLE on a single GPU. As a baseline, we consider vanilla A2C, using 120 OpenAI Gym CPU environments that send training data to the GPU to update the DNN every $N = 5$ steps. This configuration takes, on average, 21.2 minutes (and 5.5M training frames) to reach a score of 18 for Pong and 16.6 minutes (4.0M training frames) for a score of 1,500 on Ms-Pacman (Fig. 5, red line; first column of Table 3). CuLE with 1,200 environments generates approximately $2.5\times$ more FPS compared to OpenAI Gym, but this alone is not sufficient to improve the convergence speed (blue line, Fig. 5). CuLE generates larger batches but, because FPS / environment is lower when compared to CPU emulation, fewer Updates Per Second (UPS) are performed for training the DNN (Table 3), which is detrimental for learning.

**A2C+V-trace and batching strategy** To better leverage CuLE, and similar in spirit to the approach in IMPALA [6], we employ a different batching strategy on the GPU, but training data are read in batches to update the DNN every Steps Per Update (SPU) steps. This batching strategy significantly increases the DNN's UPS at the cost of a slight decrease in FPS (second columns of OpenAI Gym and CuLE in Table 3), due to the fact that the GPU has to dedicate more time to training. Furthermore, as only the most recent data in a batch are generated with the current policy, we use V-trace [6] for off-policy correction. The net result is an increase of the overall training time when 120 OpenAI Gym CPU environments are used, as this configuration pays for the increased training and communication overhead, while the smaller batch size increases the variance in the estimate of the value function and leads to noisy DNN updates (second column in Table 3, orange lines in Fig. 5). Since CuLE does not suffer from the same computational bottlenecks, and at the same time benefits from the variance reduction associated with the large number (1,200) of environments, using the same batching strategy with CuLE reduces the time to reach a score of 18 for Pong and 1,500 for Pacman respectively to 5.9 and 6.9 minutes. The number of frames required to reach the same score is sometimes higher for CuLE (Table 3), which can lead to less sample efficient implementation when compared to the baseline, but the higher FPS largely compensates for this. Extending the batch size in the temporal dimension (N-steps bootstrapping, $N = 20$) increases the GPU computational load and reduces both the FPS and UPS, but it also reduces the bias in the estimate of the value function, making each DNN update more effective, and leads to an overall decrease of the wall clock training time, the fastest

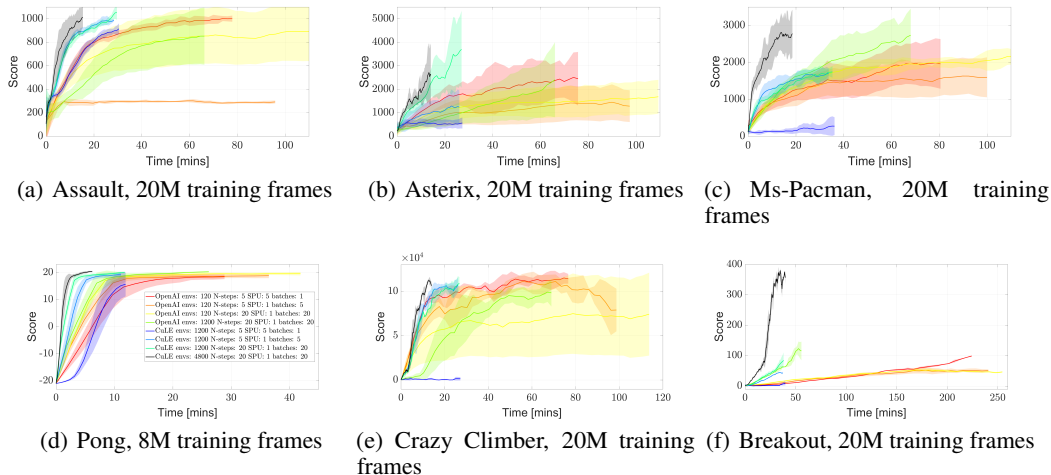

(a) Assault, 20M training frames     (b) Asterix, 20M training frames     (c) Ms-Pacman, 20M training frames

(d) Pong, 8M training frames     (e) Crazy Climber, 20M training frames     (f) Breakout, 20M training frames

Figure 5: Average testing score and standard deviation on four Atari games as a function of the training time, for A2C+V-trace, System III in Table 2, and different batching strategies (see also Table 3). Training frames are double for the multi-GPU case (black line). Training performed on CuLE or OpenAI Gym; testing performed on OpenAI Gym environments (see the last paragraph of Section 4).

convergence being achieved by CuLE with 1,200 environments. Using OpenAI Gym with the same configuration results in a longer training time, because of the lower FPS generated by CPU emulation.

**Generalization for different systems**    Table 4 reports the FPS for the implementations of vanilla DQN, A2C, and PPO, on System I and II in Table 2. The speed-up in terms of FPS provided by CuLE is consistent across different systems, different algorithms, and larger in percentage for a large number of environments. Different DRL algorithms achieve different FPS depending on the complexity and frequency of the training step on the GPU.

Table 4: Average FPS and min/max GPU utilization during training for Pong with different algorithms and using different emulation engines on different systems (see Table 2); CuLE consistently leads to higher FPS and GPU utilization.

| Algorithm | Emulation engine | FPS [GPU utilization %] | | | |
|---|---|---|---|---|---|
| | | System I [256 envs] | System I [1024 envs] | System II [256 envs] | System II [1024 envs] |
| DQN | OpenAI | 6.4K [15-42%] | 8.4K [0-69%] | 10.8K [26-32%] | 21.2K [28-75%] |
| | CuLE$_{CPU}$ | 7.2K [16-43%] | 8.6K [0-72%] | 6.8K [17-25%] | 20.8K [8-21%] |
| | CuLE | 14.4K [16-99%] | 25.6K [17-99%] | 11.2K [48-62%] | 33.2K [57-77%] |
| A2C | OpenAI | 12.8K [2-15%] | 15.2K [0-43%] | 24.4K [5-23%] | 30.4K [3-45%] |
| | CuLE$_{CPU}$ | 10.4K [2-15%] | 14.2K [0-43%] | 12.8K [1-18%] | 25.6K [3-47%] |
| | CuLE | 19.6K [97-98%] | 51K [98-100%] | 23.2K [97-98%] | 48.0K [98-99%] |
| PPO | OpenAI | 12K [3-99%] | 10.6K [0-96%] | 16.0K [4-33%] | 19.2K [4-62%] |
| | CuLE$_{CPU}$ | 10K [2-99%] | 10.2K [0-96%] | 9.2K [2-28%] | 18.4K [3-61%] |
| | CuLE | 14K [95-99%] | 36K [95-100%] | 14.4K [43-98%] | 28.0K [45-99%] |

# 5   Conclusion

As already shown by others in the case of DRL on distributed system, our experiments show that proper batching coupled with a slight off-policy gradient policy algorithm can significantly accelerate the wall clock convergence time. Although the wall clock metric only partially describes the efficiency of the learning procedure, it is among the most important from the practical point of view [18, 21, 5]; furthermore, since research is often limited by the turnaround time of experiments, reducing training

time accelerates the discovery of new algorithms. CuLE has the additional advantage of allowing effective scaling to systems with multiple GPUs.

CuLE dramatically increases the number of parallel environments, but because of the low number of instructions per second per thread on the GPU, training data can be narrow in the time direction. This can be problematic for problems with sparse temporal rewards, but rather than considering this as a pure limitation of CuLE, we believe that this peculiarity opens the door to new interesting research questions, like active sampling of important states [7, 22] that can then be effectively analyzed on a large number of parallel environments with CuLE. CuLE also hits a new obstacle, which is the limited amount of DRAM available on the GPU; studying new compression schemes, like the one proposed in [7], as well as training methods with smaller memory footprints may help extend the utility of CuLE to even larger environment counts, and design better GPU-based simulator for RL in the future. Since these are only two of the possible research directions for which CuLE is an effective investigation instrument, CuLE comes with a python interface that allows easy experimentation and is freely available to any researcher at `https://github.com/NVlabs/cule`.

## 6    Impact Statement

As interest in deep reinforcement learning has grown so has the computational requirements for researchers in this field. However, the reliance of DRL on the CPU, especially for environment simulation/emulation, severely limits the utilization of the computational resources typically accessible to DL researchers, specifically GPUs. Though Atari is a specialized DRL environment, it is arguably one of the most studied in recent times and provides access to several training environments with various levels of difficulty. The development and testing of DRL using Atari games remains a relevant and significant step toward more efficient algorithms. There are two impact points for CuLE: 1) Provide access to an accelerated training environment to researchers with limited computational capabilities. 2) Facilitate research in novel directions that explore thousands of agents without requiring access to a distributed system with hundreds of CPU cores. Although leaving RL environments "as-is" on CPUs and parallelizing across multiple nodes is indeed the shortest path to make progres, it is also inherently inefficient, in terms of the resource utilization on the local machine, and expensive, since it requires access to a large number of distributed machines. The more efficient use of the computational resources could also lead to a smaller carbon footprint.

## Footnotes

[2]Raw frames are reported here and in the rest of the paper, unless otherwise specified. These are the frames that are actually emulated, but only 25% of them are rendered and used for training. Training frames are obtained by dividing the raw frames by 4—see also [6].

[3]Other games for which we observe a similar behavior are reported in the Appendix, for sake of space.

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
