[Supplementary Material]

## 0.1 Reinforcement Learning, A2C and V-trace

**Reinforcement learning**    In RL, an agent observes a state $s_t$ at time $t$ and follows a policy $\pi = \pi(s_t)$ to select an action $a_t$; the agent also receives a scalar reward $r_t$ from the environment. The goal of RL is to optimize $\pi$ such that the sum of the expected rewards is maximized.

In model-free policy gradient methods $\pi(a_t|s_t; \theta)$ is the output of a policy DNN with weights $\theta$, and represents the probability of selecting action $a_t$ in the state $s_t$. Updates to the DNN are generally aligned in the direction of the gradient of $E[R_t]$, where $R_t = \sum_{i=0}^{\infty} \gamma^i r_{t+i}$ is the discounted reward from time $t$, with discount factor $\gamma \in (0, 1]$ (see also REINFORCE [5]) The vanilla implementation updates $\theta$ along $\nabla_\theta \log \pi(a_t|s_t; \theta) R_t$, which is an unbiased estimator of $\nabla_\theta E[R_t]$. The training procedure can be improved by reducing the variance of the estimator by subtracting a learned *baseline* $b_t(s_t)$ and using the gradient $\nabla_\theta \log \pi(a_t|s_t; \theta)[R_t - b_t(s_t)]$. One common baseline is the value function $V^\pi(s_t) = E[R_t|s_t]$, which is the expected return for the policy $\pi$ starting from $s_t$. The policy $\pi$ and the baseline $b_t$ can be viewed as *actor* and *critic* in an actor-critic architecture [4].

**A2C**    A2C [3] is the synchronous version of A3C [2], a successful actor-critic algorithm, where a single DNN outputs a softmax layer for the policy $\pi(a_t|s_t; \theta)$, and a linear layer for $V(s_t; \theta)$. In A2C, multiple agents perform simultaneous steps on a set of parallel environments, while the DNN is updated every $t_{max}$ actions using the experiences collected by all the agents in the last $t_{max}$ steps. This means that the variance of the critic $V(s_t; \theta)$ is reduced (at the price of an increase in the bias) by $N$-step bootstrapping, with $N = t_{max}$. The cost function for the policy is then:

$$\log \pi(a_t|s_t; \theta) \left[ \tilde{R}_t - V(s_t; \theta_t) \right] + \beta H[\pi(s_t; \theta)], \tag{1}$$

where $\theta_t$ are the DNN weights $\theta$ at time $t$, $\tilde{R}_t = \sum_{i=0}^{k-1} \gamma^i r_{t+i} + \gamma^k V(s_{t+k}; \theta_t)$ is the bootstrapped discounted reward from $t$ to $t+k$ and $k$ is upper-bounded by $t_{max}$, and $H[\pi(s_t; \theta)]$ is an entropy term that favors exploration, weighted by the hyper-parameter $\beta$. The cost function for the estimated value function is:

$$\left[ \tilde{R}_t - V(s_t; \theta) \right]^2, \tag{2}$$

which uses, again, the bootstrapped estimate $\tilde{R}_t$. Gradients $\nabla \theta$ are collected from both of the cost functions; standard optimizers, such as Adam or RMSProp, can be used for optimization.

**V-trace**    In the case where there is a large number of environments, such as in CuLE or IMPALA [1], the synchronous nature of A2C become detrimental for the learning speed, as one should wait for all the environments to complete $t_{max}$ steps before computing a single DNN update. Faster convergence is achieved (both in our paper and in [1]) by desynchronizing data generation and DNN updates, which in practice means sampling a subset of experiences generated by the agents, and updating the policy using an approximate gradient, which makes the algorithm slightly off-policy.

To correct for the off-policy nature of the data, that may lead to inefficiency or, even worse, instabilities, in the training process, V-trace is introduced in [1]. In summary, the aim of off-policy correction is to give less weight to experiences that have been generated with policy $\mu$, called the *behaviour policy*, when it differs from the *target policy*, $\pi$; for a more principled explanation we remand the curios reader to [1].

For a set of experiences collected from time $t = t_0$ to time $t = t_0 + N$ following some policy $\mu$, the $N$-steps V-trace target for $V(s_{t_0}; \theta)$ is defined as:

$$v_{t_0} = V(s_{t_0}; \theta) + \sum_{t=t_0}^{t_0+N-1} \gamma^{t-t_0} \left( \prod_{i=t_0}^{t-1} c_i \right) \delta_t V, \tag{3}$$

$$\delta_t V = \rho_t \big( r_t + \gamma V(s_{t+1}; \theta) - V(s_t; \theta) \big) \tag{4}$$

$$\rho_t = \min \big( \bar{\rho}, \frac{\pi(a_t|s_t)}{\mu(a_t|s_t)} \big) \tag{5}$$

$$c_i = \min \big( \bar{c}, \frac{\pi(a_i|s_i)}{\mu(a_i|s_i)} \big); \tag{6}$$

$\rho_t$ and $c_i$ are truncated importance sampling (IS) weights, and $\prod_{i=t_0}^{t-1} c_i = 1$ for $s = t$, and $\bar{\rho} \geq \bar{c}$. Notice that, when we adopt the proposed multi-batching strategy, there are multiple behaviour policies

Figure 1: FPS as a function of the environment step, measured on System I in Table **??** for *emulation only* on four Atari games, 512 environments, for CuLE$_{\text{CPU}}$; each panel also shows the number of resetting environments. A peak in the FPS at the beginning of the emulation period, as in the case of GPU emulation in Fig. **??**, is not visible in this case.

41  $\mu$ that have been followed to generate the training data — e.g., N different policies are used when
42  SPU=1 in Fig. **??**. Eqs. 5-6 do not need to be changed in this case, but we have to store all the
43  $\mu(a_i|s_i)$ in the training buffer to compute the, V-trace corrected, DNN update. In our implementation,
44  we compute the V-trace update recursively as:

$$v_t = V(s_t; \theta) + \delta_t V + \gamma c_s \big(v_{t+1} - V(s_{t+1}; \theta)\big). \tag{7}$$

45  At training time $t$, we update $\theta$ with respect to the value output, $v_s$, given by:

$$\big(v_t - V(s_t; \theta)\big)\nabla_\theta V(s_t; \theta), \tag{8}$$

46  whereas the policy gradient is given by:

$$\rho_t \nabla_\omega \log \pi_\omega(a_s|s_t)\big(r_t + \gamma v_{t+1} - V(s_t; \theta)\big). \tag{9}$$

47  An entropy regularization term that favors exploration and prevents premature convergence (as in
48  Eq. 1) is also added.

## 0.2   Thread divergence is not present in the case of CPU emulation

51  We show here that thread divergence, that affects GPU-based emulation (see Fig. **??**), does not affect
52  CPU-based emulation. Fig. 1 shows the FPS on four Atari games where all the environments share
53  the same initial state. In constrast with GPU emulation, the CPU FPS do not peak at the beginning of
54  the emulation period, where many environments are correlated.

## 0.3   Performance during training - other games

56  For sake of space, we only report (Fig. 2) the FPS measured on system I in Table **??** for three
57  additional games, as a function of different load conditions and number of environments.

58

## 0.4   Correctness of the implementation

60  To demonstrate the correctness of our implementation, and thus that policies learned with CuLE
61  generalize to the same game emulated by OpenAI Gym, we report in Fig. 3 the average scores
62  achieved in testing, while training an agent with with A2C+V-trace and CuLE. The testing scores
63  measured on CuLE$_{\text{CPU}}$ and OpenAI Gym environments do not show any relevant statistical difference,
64  even for the case of Ms-Pacman, where the variability of the scores is higher because of the nature of
65  the game.

(a) Pong       (b) MsPacman       (c) Asterix

Figure 2: FPS generated by different emulation engines on System I in Table **??** for different Atari games, as a function of the number of environments, and different load conditions (the main A2C [3] loop is run here, with N-step bootstrapping, $N = 5$.

(a) Assault       (b) Asterix       (c) Ms-Pacman       (d) Pong

Figure 3: Average testing scores measured on 10 CuLE$_{CPU}$ and OpenAI Gym environments, while training with A2C+V-trace and CuLE, as a function of the training frames; 250 environments are used for Ms-Pacman, given its higher variability. The shaded area represents 2 standard deviations.