[Reviews · NeurIPS 2020]

Review 1

Summary and Contributions: Edit: I'm raising my score from a 4 to a 5. The author rebuttal addressed a misconception I had about CuLE not being compatible with the gym interface. It seems this is not the case, which significantly improves the usability of the artifact. The authors introduce CuLE, a port of Atari to the GPU for improved throughput without needing many CPUs. Beyond the simulator artifact, the claimed contributions are: 1. identification of computational bottlenecks 2. proposal of a batching strategy for speeding up CuLE training 3. analysis of limitations of CUDA acceleration for Atari

Strengths: The throughput numbers of CuLE are impressive, as is the engineering required to accelerate Atari with CUDA.

Weaknesses: My main concern with this work is the limited scope of impact. CuLE only accelerates Atari environments. It also seems like CuLE also requires special integration with the RL libraries, which would require more work on the authors of those libraries. (1) In terms of impact, CuLE falls short of comparable work such as SampleFactory, which appears to not only have higher performance but is a more general purpose system that supports CPU-hosted environments. Similarly, Accelerated Methods for Deep Reinforcement Learning does not require special-purpose CUDA implementations for the environment. (2) Performance wise, the numbers provided by CuLE are strong but there aren't clear improvements over existing systems. For example, solving Pong in 5.9 minutes has been achieved by several RL libraries. There is also a strong focus evaluation of inference-only speed, which is quite easy to achieve in any parallel system. A strong evaluation section would compare training speed against other distributed / GPU-based RL systems. (3) The contributions beyond the CuLE artifact are limited. I did not see much novel insights in the proposed batching strategy or analysis of algorithm bottlenecks.

Correctness: Yes.

Clarity: The paper is generally well organized and clearly written. However, I do think the evaluation section could be better structured to point out takeaways from the experiments.

Relation to Prior Work: Yes.

Reproducibility: Yes

Additional Feedback:


Review 2

Summary and Contributions: Edit: I appreciate that the authors are planning to include additional experiments for more atari levels. Keeping my rating at for stated reasons. The paper implements an ATARI simulator in CUDA and demonstrates that environments can hence be run at much higher FPS using a GPU. They then run A2C and V-trace with large batches to obtain very fast training in wall clock time. As the setup is entirely on GPUs it is fairly on-policy and they can employ 1024 environments in parallel.

Strengths: Porting the Atari emulator to CUDA is a strong engineering feat and can have a large impact on the community by simplifying and accelerating atari experiments.

Weaknesses: To utilize this work large batches seem required, which limits the selection of applicable algorithms. Also I am wondering why only 4 atari games were tested. Otherwise it needs to be state that this is more an engineering contribution than a research contribution. The meta reviewer needs to decide if it fits the scope of the conference.

Correctness: The empirical approach is sound.

Clarity: Yes.

Relation to Prior Work: Yes

Reproducibility: Yes

Additional Feedback: I am assuming that the code that is promised to be submitted will ensure reproducibility of the results.


Review 3

Summary and Contributions: ===Edit after rebuttal=== After reading the author rebuttle and discussing with other reviewers, overall I have no objection in accepting, given that we all agree this would be a great tool for the RL community. However, I am keeping my score at 6 since I still see that this particular implementation would not take effect without having to execute a large number of environment (I had a misunderstanding that this would require more GPUs but the authors clarified that a single GPU is capable of executing more environment, this was helpful). The question here would be, is it really necessary that we *always* run thousands of environments? As I've pointed out in the initial review, sometimes less number of environment is sufficient enough to get a certain performance and in this situation, more environment would just be a waste of frames in my opinion. To re-iterate my review regarding "time efficiency" vs. "data efficiency", this paper addresses the former only, which is a weakness. That said, I'm okay with this as long as the authors have emphasized that this work is not aming at the latter (also in rebuttal). === The authors introduce a full-GPU supported Atari environment, called CuLE, and perform a thorough analysis of the improvement over CPU implementations. The analyses and comparisons to the commonly used OpenAI Gym environment are very detailed, with good insights into when is one tool better/worse than others. Open-sourcing this tool (after the review period) would make a good contribution to RL research platforms.

Strengths: The part that I like the most about this paper is that the authors recognized/disentangled different factors when considering "speed". The emulator speed, which is the part that is usually implemented in CPU; this is what the authors' main contribution here, to implement the emulator in GPU. The inference speed, which is the communication cost between CPU-GPU; this is what can be improved by leveraging CuLE to reduce this cost. And the training speed, which is the limitation of an RL algorithm; this part was not addressed in the paper (i.e., the authors used an existing algorithm to test CuLE). Having a clear definition of each component inside a training system is very helpful in identifying the bottlenecks. As shown in the experiments, the throughput of an emulator itself can be one of the reasons why some games run slower than others. I also appreciate the "generalization for different systems" experiments, which shows that CuLE will be a good tool in most of the commonly used RL algorithms. The comparison with existing acceleration methods in Table 1 also provides a clear view of the advantage of CuLE.

Weaknesses: My main concern is that results seem to be contradictory to what the authors claimed as the benefit of leveraging GPU accelerations. Specifically, in the "impact statement" the authors described CuLE can "provide access to an accelerated training environment to researchers with limited computational capabilities," but the results show the acceleration won't take into effect unless you use more computation---Figure 2, CuLE runs slower than OpenAI when using a fewer number of environments. If someone can only afford to run 100 environments, would this mean CuLE is not useful here? The limitation of the memory has been noted in the paper which is good. I was confused when looking at Table 3. First, why is there no 120 envs experiment for CuLE? Second, by looking at the columns of OpenAI and the columns of CuLE, the message I get was "CuLE has a higher FPS, but somehow performs less number of UPS, and it takes more data and longer time to achieve the same performance as OpenAI that uses a fewer number of envs." Take Assault as an example, if I can achieve 800 scores using 120 OpenAI CPU envs within 5M frames, why should I bother using CuLE with 4 GPUs and consume 18M frames? Of course, this depends on what you are trying to optimize: CuLE optimizes wall-clock time (when you have the computation resources), but often when I consider speed up RL, I care more about being able to learn with less number of environmental interactions (i.e., frames), because collecting data is often more expensive than computation (e.g., in robotics it is impossible to let a robot repeat a movement for millions of times). I hope the authors can elaborate further on this. This might be a point worth addressing in the impact statement: CuLE might create a resource imbalance problem among researchers; big companies can easily leverage CuLE to produce results faster and better, while small labs will be unable to do so, result in slower progress.

Correctness: All methods are sound.

Clarity: The paper is well written and easy to read.

Relation to Prior Work: A clear comparison with existing acceleration methods is made in Table 1 of the paper

Reproducibility: Yes

Additional Feedback: Some minor formatting issues: -no need to insert a line break between section titles & text -reference [23] and [24] are duplicated -line 127 "...TIA kernel may be scheduled *one* the GPU with more than one thread per game..." -> *on* the GPU? -there are several undefined references to Figures and Tables inside the supplementary materials -the impact section should be titled "broader impact" instead of "impact statement" Overall, I think this paper presents a good open-source tool for the RL community. And as the authors discussed, CuLE can open up new directions in designing RL algorithms that can fully leverage the hardware advantage (i.e., improve the "training speed"). I wonder if the authors have a plan to extend their implementation to more environments (e.g., can you make a full wrapper around all OpenAI gym environments), which will certainly have a great impact.


Review 4

Summary and Contributions: The authors proposed CuLE, a CUDA port of the Atari Learning Environment (ALE) CuLE utilizes GPU parallelization to render frames directly on the GPU which removes the current inference bottleneck for Atari in RL loop: CPU-GPU communication. This also enables running thousands of games simultaneously on a single GPU. Given the fact that Atari is still one of the most popular environments for RL research, this paper can potentially be extremely high impact, assuming the authors also make it easy to use! CuLE: 1) Provides access to an accelerated training environment to researchers with limited computational capabilities. 2) Facilitate research in novel directions that explore thousands of agents without requiring access to a distributed system with hundreds of CPU cores

Strengths: The paper analysis the computational bottleneck of the current methods in a systematic way and addresses them by porting the Atari emulation into the GPU. There are great discussions on how this port could be done and the authors clearly justify the reasoning on their decisions. Finally, there are well designed experiments that illustrate the achieved improvements with clarity.

Weaknesses: This is not a weakness of the paper per se but a way that the authors could improve the paper substantially to make it even stronger for machine learning conference: The paper provides a method for emulating Atari faster, but the authors do not demonstrate how this can be used to achieve high scores, using the same method, in a shorter amount of time. There is Figure 5, but what I have in mind is a table with scores across many games by training the same model for a fixed amount of time. This can show the strong impact that this paper can have in shortening the scientific experiment cycle.

Correctness: Yes

Clarity: The paper is very well written and easy to follow. This paper is not in my direct field of expertise and I was not familiar with some of the concepts but I did manage to follow and understand. However, I encourage the writers to rewrite some parts of the paper to make it more "NeurIPS friendly", given that NeurIPS is not a system conference. e.g. the reads can use a one liner description of what TIA is! I also encourage the authors to include a table of abbreviations given that there are too many of them e.g. SPU, UPS, etc. The clarity of some of the graphs can be improved .e.g. the labels in Fig.5 are really tiny.

Relation to Prior Work: The authors explored the previous deep RL methods in depth and recognized their computational bottleneck. This paper is a direct answer to this bottleneck which, in theory, should improve the majority of the previous methods.

Reproducibility: Yes

Additional Feedback:

[Author Response · NeurIPS 2020]

Based on the interesting reviewers' comments, we believe that it is important to better clarify here (and, upon acceptance, in the final version of the paper) our two major contributions.

The **first main contribution** is CuLE, a "strong engineering feat (R2)" which achieves "impressive throughput" (R1) in the simulation of Atari games, which "can have a large impact on the community by simplifying and accelerating Atari experiments: (R2)", especially if "the authors also make it easy to use (R4)". This is indeed the case, as the released version of CuLE will be characterized by a python interface which is fully compatible with OpenAI Gym, and apart from installing it, CuLE will not "require any special integration with the RL libraries (R1)", a potential (but fortunately non-existing) drawback indicated by R1.

R1 suggests that "there aren't clear improvements over existing systems... There is also a strong focus on the evaluation of inference-only speed, which is quite easy... in any parallel system. A strong evaluation section would compare training speed against other distributed / GPU-based RL systems". We believe that Table I actually contains such comparison, but we prefer reading it from the point of view of R4: CuLE was not designed to achieve the highest possible throughput when compared to large/costly distributed systems, but to "provide access to an accelerated training environment to researchers with limited computational capabilities" and "facilitate research in novel directions that explore thousands of agents without requiring access to a distributed system with hundreds of CPUs". Indeed, the point that we want to make is that, with a system including 1 or at most 4 GPUs, CuLE's throughput is of the same order of magnitude of that of large (and much more expensive) distributed systems, like IMPALA [7] or a DGX-1 [23].

Regarding R2's comment that "the acceleration won't take into effect unless you use more computation... CuLE runs slower than OpenAI when using a fewer number of environments", we want to highlight once more that CuLE leverages at best the power of the already-available computational resources by achieving high GPU occupancy and utilization. The cost (in dollars) for scaling from hundreds of CPU/GPU environments to thousands of GPU-CuLE environments is virtually null, as at least one GPU is likely to be present in any system used for RL. Although it remains true that "big companies can easily leverage CuLE to produce results faster and better (R2)", we disagree with the additional assumption that "small labs will be unable to do so, resulting in slower progress (R2)": we hope that CuLE can accelerate the workflow of single researchers even if using one GPU, resulting in faster progress, while the practical advantage of scaling to hundreds of thousands of environments still have to be demonstrated, especially for Atari.

The **second main contribution** is related to the "thorough analysis of the improvement over CPU implementations (R3)" and the provided insights, that are not simply intended to give an "analysis of the algorithm bottleneck (R2)". Instead, based on the CuLE experience and by "recognizing/disentangling different factors when considering "speed" (R3)", e.g. by analyzing the processes of data generation, transmission, storage, and consumption, we identify system level bottlenecks that have a negative impact on the performance of not only CuLE, but most parts of the existing RL frameworks, and consequently derive design principles that we applied in CuLE but can be easily generalized to design and implement effective RL training systems (including "more OpenAI Gym environments (R3)") that make use of the available computational resources at best, especially when running on GPUs. To give an example of how these insights may be useful for the research community, R2 notices that "large batches seem required, which limits the selection of applicable algorithms". More than interpreting this as a simple limitation of CuLE, we believe that it may be a peculiarity of any large throughput simulation system running on GPU, that is at the same time penalized in terms of frames per second per environment. We believe that porting this (and similar) observation to the attention of the researchers can only be beneficial for the design of future simulation libraries and to develop RL algorithms that leverage different learning paradigms, depending on the specific data generation pattern. Strictly related to this topic is the observation of R3 about the sample efficiency of A2C+V-Trace with CuLE: "if I can achieve 800 scores using 120 OpenAI CPU envs within 5M frames, why should I bother using CuLE with 4 GPUs and consume 18M frames?". To answer this, we make reference to the abundant recent literature on Evolutionary Strategies for RL (e.g. see Evolution Strategies as a Scalable Alternative to Reinforcement Learning, 2017, just to mention one), highlighting that wall-clock time may be more important than sample efficiency from the practical point of view. At the same time we want to highlight once more that the sample inefficiency observed in our paper (Table III) is probably associated to the data generation / consumption pattern, and thus deserves more attention in future research to be better understood. We believe that CuLE can not only trigger, but also facilitate and speed-up this kind of research activity.

Finally, we find very interesting that R1 mentioned the Sample Factory paper in his review — this paper was published on Arxiv after our submission, nevertheless it is based on a system level analysis which is similar in spirit to ours (and can be summarized as "find and remove the bottleneck at system level"), but based on a completely different implementation based on CPU simulation and an RL algorithm using an asynchronous sampler. Assuming that this paper may have been submitted to NeurIPS as well, we believe it may be very instructive to compare the different design approaches and combine the best aspects of our and their solution.

A last note, we can add more training examples / curves in the additional material upon paper acceptance, as requested by R2.

[Meta-Review · NeurIPS 2020]

There was a consensus by the reviewers that this paper should be accepted. The paper provides a Cuda implementation of the ATARI simulator which allows for the running of RL experiments on GPUs. This is a solid contribution and has the clear potential to literally accelerate reinforcement learning research.